# From Atomic Level to Large-Scale Monte Carlo Magnetic Simulations

**DOI:** 10.3390/ma13173696

**Published:** 2020-08-21

**Authors:** Artur Chrobak, Grzegorz Ziółkowski, Dariusz Chrobak, Grażyna Chełkowska

**Affiliations:** 1Institute of Physics, University of Silesia in Katowice, 75 Pułku Piechoty 1A, 41-500 Chorzów, Poland; grzegorz.ziolkowski@us.edu.pl (G.Z.); grazyna.chelkowska@us.edu.pl (G.C.); 2Institute of Materials Engineering, University of Silesia in Katowice, 75 Pułku Piechoty 1A, 41-500 Chorzów, Poland; dariusz.chrobak@us.edu.pl

**Keywords:** Monte Carlo simulations, magnetic simulations, magnetic materials

## Abstract

This paper refers to Monte Carlo magnetic simulations for large-scale systems. We propose scaling rules to facilitate analysis of mesoscopic objects using a relatively small amount of system nodes. In our model, each node represents a volume defined by an enlargement factor. As a consequence of this approach, the parameters describing magnetic interactions on the atomic level should also be re-scaled, taking into account the detailed thermodynamic balance as well as energetic equivalence between the real and re-scaled systems. Accuracy and efficiency of the model have been depicted through analysis of the size effects of magnetic moment configuration for various characteristic objects. As shown, the proposed scaling rules, applied to the disorder-based cluster Monte Carlo algorithm, can be considered suitable tools for designing new magnetic materials and a way to include low-level or first principle calculations in finite element Monte Carlo magnetic simulations.

## 1. Introduction

Simulations of magnetization processes have significant meaning in regards to both scientific and practical points of view [1,2,3,4,5,6]. The observed progress in technologies utilizing magnetic materials requires new magnets with unique properties optimized for different applications. What is more, a permanent demand for soft and hard magnets with ultimate characteristics can be observed [7,8,9,10,11]. Designing such systems should include modeling of the magnetization processes using computer simulations, which enables the searching for and testing of their properties in a pre-lab state. In this field, two main directions can be indicated, i.e., simulations based on the Landau equation [12,13,14] or Monte Carlo (MC) rules [15,16,17,18,19]. Both methods have advantages and disadvantages depending on the intended use of the considered magnetic system. The first approach is rather dedicated to continuous large-scale systems in which phenomena occurring at the atomic level are represented by some average volumetric parameters. On the other hand, the MC simulation methods are based on inter-atomic properties, and therefore, modeling of mesoscopic systems results in significant consumption of computational resources. This problem can be solved using parallelization of the MC algorithm [20,21]. Unfortunately, in order to study magnetization processes of ferromagnetic systems, it is necessary to utilize one of the cluster MC methods, and therefore, the possible parallelization is rather problematic. One can imagine a situation in which a node in a system represents a “super-cell” instead of single spin or magnetic moment. In this way, the analyzed system can be enlarged by means of the super-cell size. The question is how to ensure thermodynamic as well as energetic equivalence between the initial (with spins) and the enlarged (with super-cells) systems, transferring properties on the atomic level to the approach based on the finite element method.

In this work, we present the scaling rules for MC methods and their application. In particular, we propose that the rules be applied to the disorder-based cluster MC algorithm [22], formulated and developed by our team, which is a promising tool for studying and designing new magnetic materials. 

## 2. Scaling Rules 

A typical MC Metropolis algorithm refers to single spins (magnetic moments) placed in the so-called nodes with defined distances *r*. This well-known algorithm lies in a random walk over the system nodes and change of the spin direction with the acceptance probability equal to unity if it decreases the energy of the system. If the energy increases, the spin-flip can be accepted with the Metropolis probability *exp*(−Δ*E/k_B_T*) where Δ*E* = *E_new_* − *E_old_* is the difference in energy between the new and old spin configurations and *k_B_T* is the thermal energy.

Our approach involves the magnetic energy of the spin system (further called system energy) calculated in the frame of the continuous 3D Heisenberg model [23]:(1)$$E=-\underset{i,j}{\sum }J_{ij}S_{i}\cdot S_{j}-\underset{i}{\sum }K_{i}{\left({\hat{S}}_{i}\cdot {\hat{n}}_{i}\right)}^{2}-g\mu_{B}\mu_{0}\underset{i}{\sum }H_{i}\cdot S_{i}+D\underset{i,j}{\sum }\frac{S_{i}\cdot S_{j}-3\left(S_{i}\cdot e_{ij}\right)\left(S_{j}\cdot e_{ij}\right)}{r_{ij}^{3}}$$
where *J_ij_* is the exchange parameter, ***S****_i_* is the spin vector on site *i*, *K_i_* is the anisotropy constant (per site), $\hat{n_{i}}$ is the versor of the easy magnetization axis, $\hat{S_{i}}$ is the versor of the spin, *g* is the Lande factor, *μ_B_* is the Bohr magneton, *μ_0_* is the vacuum permeability, ***H****_i_* is the magnetic field on site *i*, *D* is the dipolar constant, ***e****_ij_* is the directional versor between the *i-th* and *j-th* nodes and *r_ij_* is the distance between the *i-th* and *j-th* nodes.

In order to effectively simulate magnetization processes of ferromagnetic materials, it is necessary to use the disorder-based cluster MC method that we also consider appropriate for multi-phase magnetic systems. This approach is based on the Wolff clusterization technique [17] applied to a spin continuous system (described in detail in [22]). Here we present only a general idea based on modification of the so-called adding probability (adding a spin to a cluster) by an additional factor attributed to a local configuration (information) entropy of the selected system’s property (the magnetic anisotropy in our case). Finally, the adding probability takes the form
(2)$$P_{ij}^{add}=\left(1-exp\left(\frac{-E_{ij}^{coupling}}{k_{B}T}\right)\right)exp\left(-\alpha S_{i}^{loc}\right)$$
where $E_{ij}^{coupling}$ is the direct exchange coupling energy between the spins attributed to nodes *i* and *j*, $S_{i}^{loc}$ is the local configuration entropy of anisotropy (calculated in the defined sphere around the *i-th* node) and *α* is the factor responsible for strengthening and weakening of the entropy impact on the adding probability.

It turns out that the entropy-based modification of adding probability supports and significantly streamlines simulations of magnetization processes in multiphase magnetic systems [24,25]. In order to make a profit from this method, as a tool for designing magnetic materials, it is necessary to analyze relatively large-scale systems, which has a key meaning for both the qualitative and quantitative analyses. Here, we propose the scaling rules, which enable the modeling of large-scale mesoscopic systems, with an acceptable level of computing resource consumption.

Let us consider an initial 3D system of nodes, each occupied by a single spin represented by the vector ***S***, spaced apart by *r* (refer to Figure 1a). Our re-scaled system is now composed of a set of super-cells characterized by spin vector ***S***’ and distances between them *r*’ (Figure 1b). The *r*’/*r* ratio defines an enlargement factor *n* which has a key meaning for further considerations. Each super-cell reflects the magnetic moment ***S***’ of *n*^3^ spins located in the box of dimensions *nr* × *nr* × *nr*.

In this way, the number of nodes (representing the system) can be reduced by the *n*^3^ ratio. In order to carry out MC simulations, it is necessary to keep (i) an equivalence between energies of the original and re-scaled systems when the direction of spin ***S*** or ***S***’ changes and (ii) detailed thermodynamic balance during the iteration steps.

The first condition can be fulfilled by analyzing Equation (1) and taking into account that the acceptance and adding probabilities include a change of the system energy related to the spin’s direction changes. Generally, we assume that the “spin” ***S***’, attributed to the super-cell in the re-scaled system (3D case and ferromagnetic coupling), is equal to *n*^3^***S***. In addition, all linear distances between the nodes are re-scaled by taking *r*’ = *nr*. The energy change calculated for the re-scaled system can be described using similar relations to the single-spin system but with ***S***’, *r*’ and some apparent quantities like *J*’, *K*’ and *D*’.

For the change of the exchange energy, the apparent exchange integral parameter *J_ij_*’ (energy difference between parallel and anti-parallel super-cells) should be defined in the following way:(3)$$\Delta E_{ij}^{\prime }=-2J_{ij}^{\prime }S_{1}^{\prime }S_{2}^{\prime }=-2J_{ij}^{\prime }S_{1}S_{2}n^{6}=-2J_{ij}S_{1}S_{2}n^{2}$$

The last term expresses exchange coupling between the neighboring super-cells and includes only the spins on the common wall, as shown in Figure 1b. From this equation, one can determine
(4)$$J^{\prime }=Jn^{-4}$$

Equation (1) contains the so-called dipolar energy term, for which the scaling rules should account for the dependence on the relative position between interacting spins. For a simplification of this long-range energy term, we found that the value of the D parameter can remain unchanged for the re-scaled system. The conducted (however, not shown here) comparison between energies calculated for the group of single spins ***S*** and the corresponding magnetic super-cells (with re-calculated spins ***S***’ and distances *r*’), reveals a small error of order 1%. Apart from that, for hard magnetic phases, these kinds of interactions have, generally, marginal meaning due to the high value of magnetic anisotropy energy. Nevertheless, for soft magnetic phases, the error must be taken into consideration.

In addition, the anisotropy constant should be re-scaled taking into account the *n*^3^ and *n*^2^ factors for anisotropy in the volume and on the surface, respectively. Table 1 summarizes the proposed scaling rules for all parameters appearing in the expression of system energy Equation (1).

The proposed scaling rules should also ensure a detailed thermodynamic balance, similar to the single-spin system. In the MC type of simulation, such a balance is realized through the proper choice of the adding probability, which is a kind of competition between energy change (caused by MC step) and thermal energy *k_B_T*. For the re-scaled system, the MC step means a collective change of all spins in a chosen super-cell. It is worth writing an explicit Δ*E*’ with Equation (1) using the scaling rules
(5)$$\begin{array}{ll} \Delta E_{i}^{\prime } & =-\underset{j}{\sum }{J^{\prime }}_{ij}\Delta {S^{\prime }}_{i}\cdot {S^{\prime }}_{j}-{K^{\prime }}_{i}{\left(\Delta \hat{S_{i}^{\prime }}\cdot \hat{n_{i}}\right)}^{2}-g\mu_{B}\mu_{0}H_{i}\cdot \Delta {S^{\prime }}_{i}+D\underset{j}{\sum }\frac{\Delta {S^{\prime }}_{i}\cdot {S^{\prime }}_{j}-3\left(\Delta {S^{\prime }}_{i}\cdot e_{ij}\right)\left({S^{\prime }}_{j}\cdot e_{ij}\right)}{{r^{\prime }}_{ij}^{3}}\\ & =-n^{2}\underset{j}{\sum }J_{ij}\Delta S_{i}\cdot S_{j}-n^{3}K_{i}{\left(\Delta \hat{S_{i}}\cdot \hat{n_{i}}\right)}^{2}-n^{3}g\mu_{B}\mu_{0}H_{i}\cdot \Delta S_{i}+n^{3}D\underset{j}{\sum }\frac{\Delta S_{i}\cdot S_{j}-3\left(\Delta S_{i}\cdot e_{ij}\right)\left(S_{j}\cdot e_{ij}\right)}{r_{ij}^{3}} \end{array}$$

Consequently,
(6)$$\Delta E_{i}^{\prime }=n^{3}\Delta e_{i}$$
where
(7)$$\Delta e_{i}=-n^{-1}\underset{j}{\sum }J_{ij}\Delta S_{i}\cdot S_{j}-K_{i}{\left(\Delta \hat{S_{i}}\cdot \hat{n_{i}}\right)}^{2}-g\mu_{B}\mu_{0}H_{i}\cdot \Delta S_{i}+D\underset{j}{\sum }\frac{\Delta S_{i}\cdot S_{j}-3\left(\Delta S_{i}\cdot e_{ij}\right)\left(S_{j}\cdot e_{ij}\right)}{r_{ij}^{3}}$$
is the energy change of, let us say, a reduced “e-system”, similar to the original one but with the exchange energy divided by the *n* parameter. It is interesting that the energy change of the super-cell Δ*E*’ is equivalent to the energy change of the single spin in *n*^3^ e-systems simultaneously. Through this conclusion, the re-scaled system can be simulated with the thermodynamic balance determined for the reduced one, which is more effective, taking into account the fact that *exp*(−Δ*E*’*/k_B_T*) << *exp*(−Δ*e/k_B_T*). It should be emphasized that the most important assumption is the coherent rotation of all spins in the super-cells. This causes a restriction in the temperature range, which has to be significantly lower than the Curie point. Therefore, the value *k_B_T* can be considered a kind of “annealing temperature”, enabling effective relaxation of the system and leading to minimization of its free energy.

In summary, the algorithm used and modus operandi are briefly described in the following points:

Step 1. Choose the 3D system size, fitting it to computing resources.

Step 2. Generate a structure of the system and assign the magnetic quantities (*J_ij_*, ***S**_i_*, *K_i_* and distance between the nodes *r_ij_*) to each *i-th* node. In this stage, the nodes should be considered atoms with a localized magnetic moment and the parameters should reflect their exchange coupling, magnetic moment and anisotropy.

Step 3. Define the *n* parameter as the re-scaling factor.

Step 4. Assign to each *i-th* node the re-scaled magnetic quantities (*J_ij_*’, ***S**_i_*’, *K_i_*’ and distance between the nodes *r_ij_*’) using the relations listed in Table 1.

Step 5. Define temperature *T* (or better *k_B_T* in relation to *J_ij_*) and external magnetic field ***H***.

Step 6. Relax the system by carrying out *N_relax_* simulation steps (see below).

Step 7. Calculate average values (in our case, average magnetization) by carrying out *N_avr_* simulation steps.

Step 8. Change the field ***H*** and go to 6.

The simulation step is based on the cluster MC method, described in detail in [22]. Figure 2 schematically shows the single iteration step including the cluster building procedure as the modification of the Metropolis algorithm. 

Focusing on the re-scaling rules, the most important aspect is the determination of the acceptance probability, which depends on the competition between the system energy change (calculated using Equation (5)) and the thermal energy (*k_B_T*). As was demonstrated (Equations (5)–(7)), this ratio can be calculated in the following two ways:Calculate the energy change (Δ*E*’) using the re-scaled quantities and take *exp*(−*n^−3^*Δ*E*’/*k_B_T*) as the acceptance probability.Calculate the energy change (Δ*e*) using the unscaled distances *r_ij_*, spins ***S****_i_*, anisotropy constants *K_i_* and *J_ij_*/*n* as the exchange integral parameter, taking *exp*(−Δ*e*/*k_B_T*) as the acceptance probability.

The question may arise of how a limit of the system enlargement can be defined by the *n* parameter. As the super-cell should be magnetically uniform, its size should not exceed a critical diameter for which magnetization reversal takes place by coherent rotation. More precisely, based on the system parameters (*J*, *S*) one can determine a coherence radius *R*_coh_ which is the maximum size of a uniformly magnetized particle [26]. For various magnetic materials, it ranges from more than 10 nm to even more than 1 µm for magnetically soft and hard magnets, respectively [27].

An important aspect of the proposed approach is a direct relation between low-level calculations and the finite element Monte Carlo magnetic simulations. The quantities describing the re-scaled system, e.g., spin vector, exchange integral parameter and magnetic anisotropy attributed to the super-cell, are derived from the unscaled one. In a situation when the nodes in the unscaled system represent atoms in a crystal structure, the mentioned quantities can be determined using, for example, density functional theory (DFT) based calculations.

## 3. Results and Discussion

In order to show and prove the proposed approach, systems of ferromagnetic sphere, regular box (*a* = *b* = *c*) and tetragonal box (*a* = *b*, *c* = *2a*) were simulated with the use of different values of the scaling factor *n*. The chosen calculation procedure was the same as the one presented in [22,24,25]. The system parameters were as follows: number of system nodes 40 × 40 × 40 = 64,000, *r* = 0.28 nm, *J_ij_* = 1 × 10^−2^ eV, *K_i_* = 0 (perfect soft magnet), *S_i_* = 1, *k_B_T* = 1 × 10^−5^ eV, *D* = 2.18 eVnm^3^ and *n* = 1, 10, 50 and 100. The other parameters appearing in the presented algorithm are *N_relax_* = *N_avr_* = 400, *P*_cl_ = 0.001 and *θ* = π/100. The initial 3D systems (for *n* = 1) and their magnetic states (*H* = 0) for *n* equal to 10, 50 and 100 are depicted in Figure 3, Figure 4 and Figure 5, respectively. The arrows represent magnetic moments assigned to the nodes, and color depends on the arrow direction. For all cases with *n* = 1, the magnetic moment alignment is ferromagnetic because the contribution of the dipolar energy is marginal. With the development of the system size, the appearance of different magnetic structures dependent on the *n* parameter and the shape of the objects can be observed. In the case of *n* = 50 and for all analyzed objects, the depicted configurations of magnetic moments have pure vortex (spheres) and vortex-like (boxes) characters. For the higher value of *n*, more complex magnetic structures were detected. Such magnetic structures are expected and experimentally observed in, for example, magnetically soft amorphous and nanocrystalline iron-based alloys, such as the so-called fingerprint domains.

The second demonstration of the proposed scaling rules was a simulation of magnetization processes for the tetragonal box with magnetic anisotropy (*K* = 5 × 10^−5^ eV) directed along the z-axis. Magnetic field, also applied in the z-axis, ranged from 1.2 to −1.2 T in order to obtain a full hysteresis loop. The *n* parameter was equal to 100 and the remaining parameters of the system were the same as for the previous example (see Figure 5). In this configuration, a quadratic hysteresis loop was expected and obtained using the above-described procedure. Figure 6e shows a normalized magnetization (z-component of average magnetic moment) in a function of the applied external magnetic field, as shown in the graph in the center. Additionally, magnetic moment configurations in fields around the magnetization jump are presented as a background. It can be seen that the reverse magnetization process started from a “seed” nucleated on the top surface (Figure 6a) and then the domain expanded, reaching new saturation according to the direction of the external magnetic field. The Figure 6c depicts the growing domain responsible for the rapid magnetization change. This scenario was expected due to the fact that the magnetic moments on the top (or bottom) surface are more weakly coupled in comparison to the magnetic moments in the volume. Moreover, for this irregular shape and value of *n*, the dipolar interactions tend to reverse the directions of the magnetic moments, especially near the top and bottom surfaces (see also Figure 5).

## 4. Concluding Remarks

Simulations of magnetization processes can be very useful tools for designing new magnetic materials. It is of great importance to include atomic properties in considerations for studying magnetization processes of dimensionally real systems, i.e., mesoscopic in size. The presented MC method, with a disorder-based cluster approach, is suitable for multimagnetic systems, but the system size (counted in nodes) is restricted to 10^5^–10^6^ nodes, which means that only nano-objects can be analyzed. In order to enlarge this size, we propose re-scaling of the system using a concentration of the volume *nr* × *nr* × *nr* (*n* is the scaling factor, *r* is inter-node distance) into one node. The new system contains the same number of nodes but all its linear dimensions are enlarged by the *n* factor. Simultaneously, it is necessary to recalculate the system parameters occurring in a Hamiltonian, which is used for energy change calculation. On the whole, we propose scaling rules, which save energetic equivalence between rotation of the re-scaled magnetic moment and a group of spins in the *nr* × *nr* × *nr* volume. Additionally, we have found that the MC iterations can be carried out under the thermodynamic balance determined for the “reduced” system, similar to the initial one but with an exchange integral parameter equal to *J_ij_/n*. 

The presented magnetic states for various objects with different values of the *n* parameter confirm the correctness of the proposed approach. The magnetic moment configurations revealed the transition from pure ferromagnetic to vortex or fingerprint structures, when the system size had been enlarged. The method was also shown to be useful for simulation magnetization processes of hard magnetic materials. In this case, the full hysteresis loop as well as magnetic moment configuration were determined, revealing some characteristic behaviors of the magnetic moments that are responsible for the shape of the reverse magnetization curve.

In conclusion, the proposed scaling rules facilitate simulation of large-scale systems based on the parameters determined for atomic level interactions. Our approach can be considered one way of “connecting” low-level or first principle calculations to finite element MC magnetic simulations.

## Figures and Tables

**Figure 1 materials-13-03696-f001:**
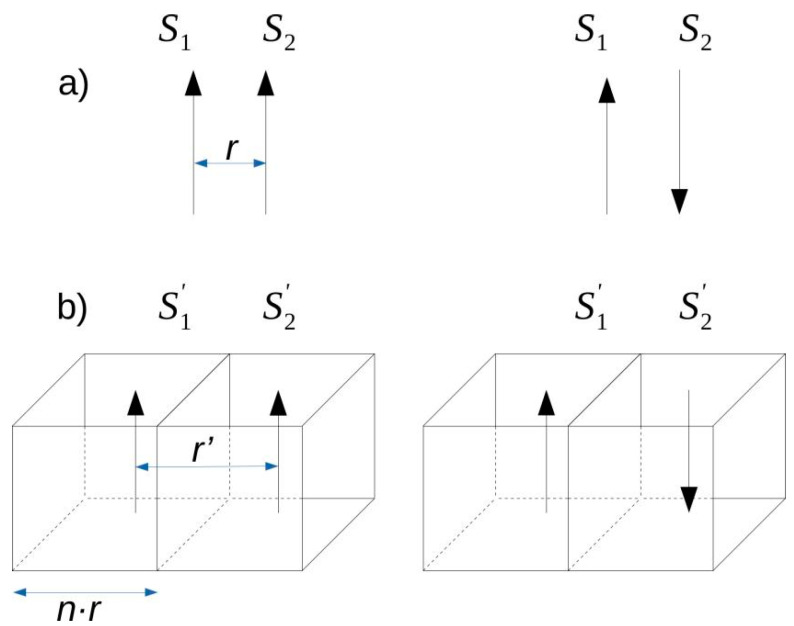
The idea of the spin system re-scaling defined by the enlargement factor *n*. (**a**) spin configuration for the original system; (**b**) spin configuration for the re-scaled system.

**Figure 2 materials-13-03696-f002:**
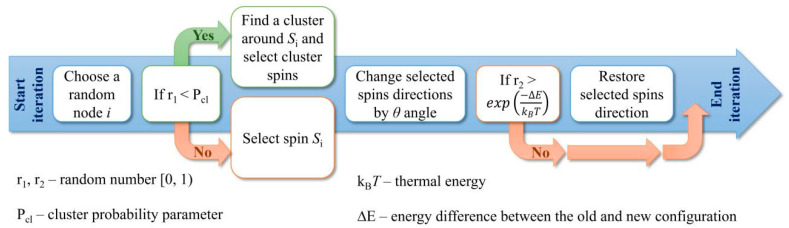
Flowchart of single iteration steps used in this paper.

**Figure 3 materials-13-03696-f003:**
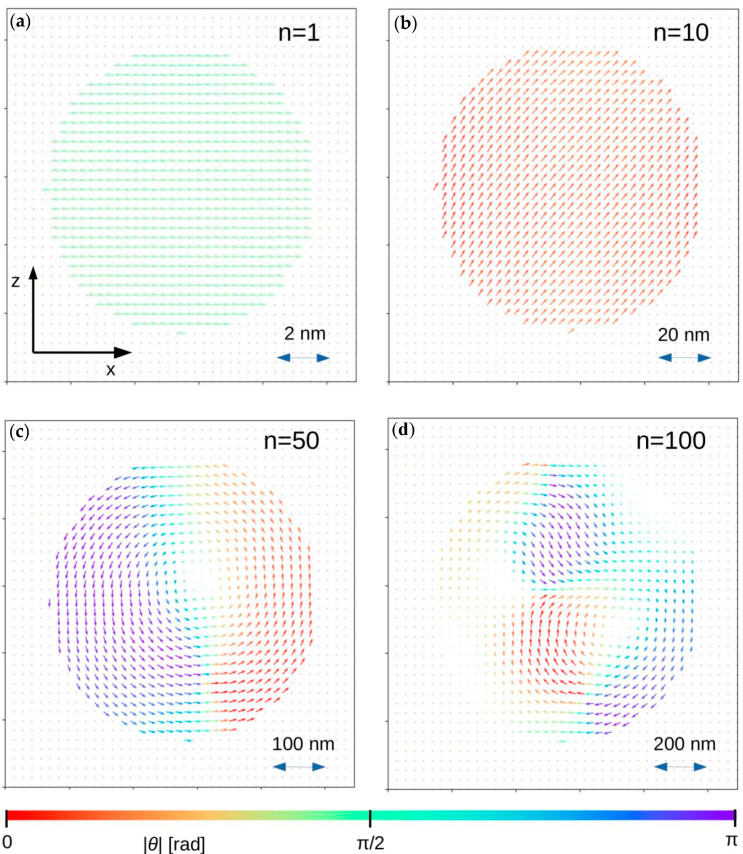
Magnetic moment configurations for 3D sphere in central z-x plane for different *n* parameters. In this figure, *θ* is the angle between magnetic moment direction and the *z-axis*. (**a**) *n* = 1; (**b**) *n* = 10; (**c**) *n* = 20; (**d**) *n* = 100.

**Figure 4 materials-13-03696-f004:**
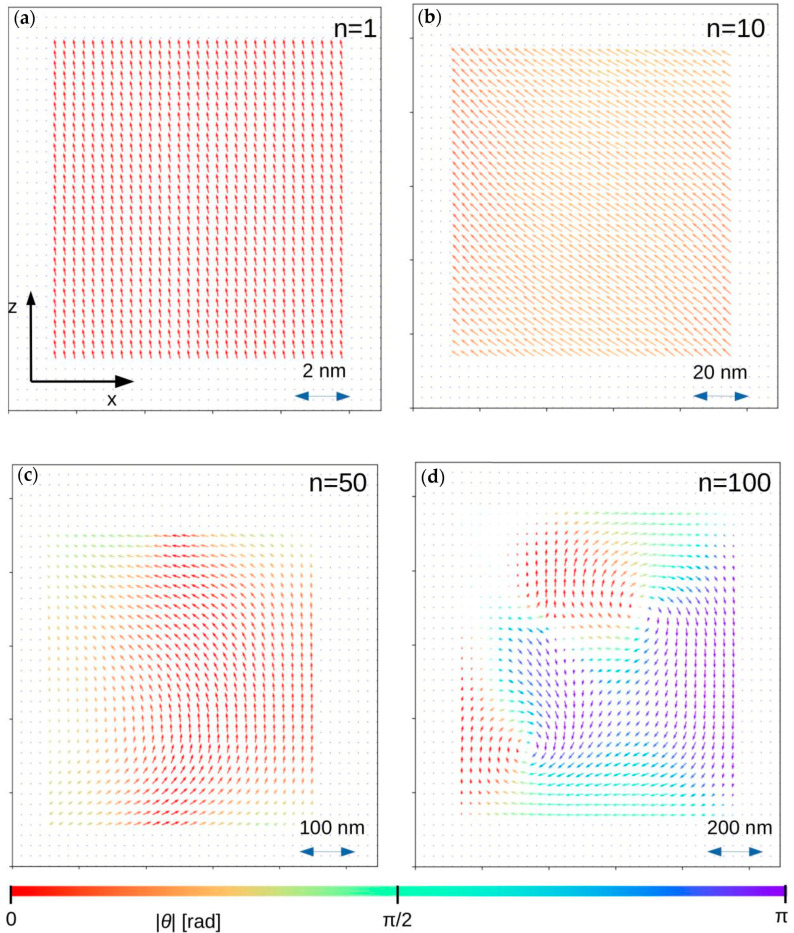
Magnetic moment configurations for 3D regular box in central z-x plane for different *n* parameters. In this figure, *θ* is the angle between magnetic moment direction and the *z-axis*. (**a**) *n* = 1; (**b**) *n* = 10; (**c**) *n* = 20; (**d**) *n* = 100.

**Figure 5 materials-13-03696-f005:**
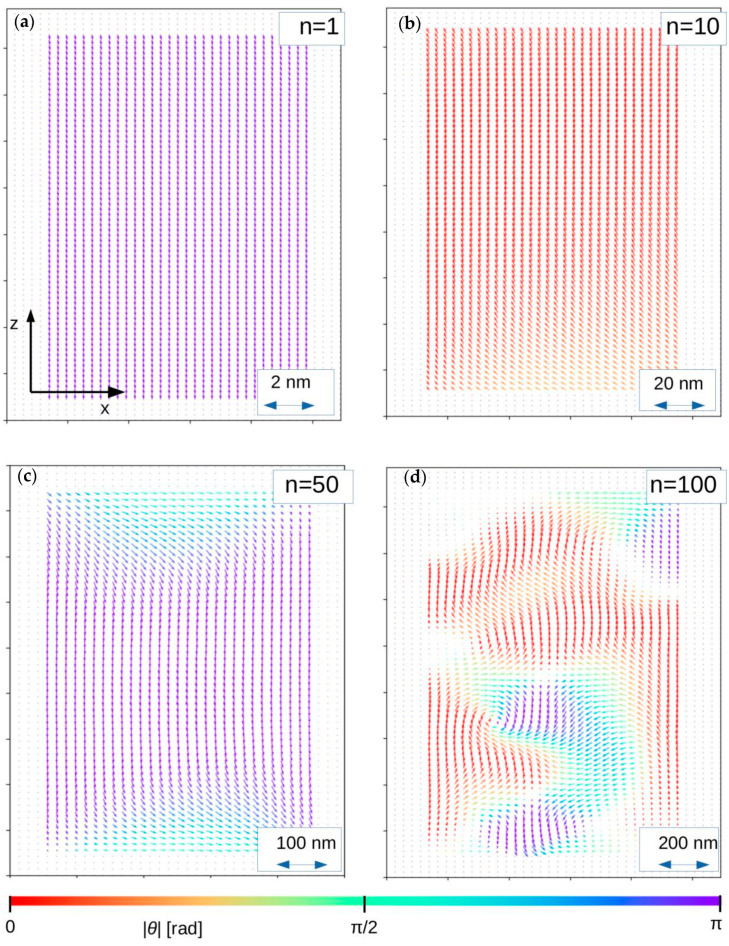
Magnetic moment configurations for 3D tetragonal box in central z-x plane for different *n* parameters. In this figure, *θ* is the angle between magnetic moment direction and the *z-axis*. (**a**) *n* = 1; (**b**) *n* = 10; (**c**) *n* = 20; (**d**) *n* = 100.

**Figure 6 materials-13-03696-f006:**
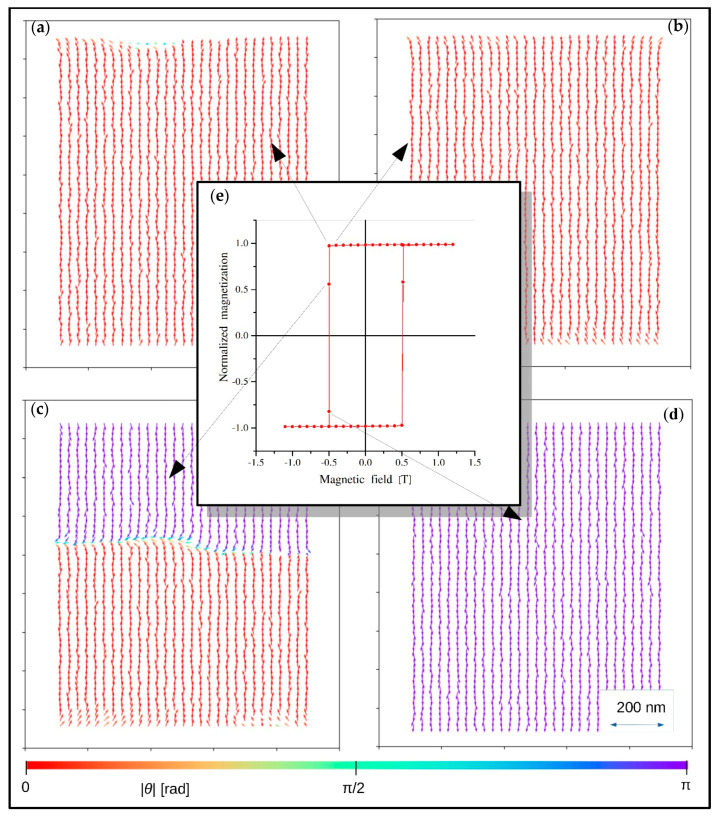
Hysteresis loop and magnetic moment configuration (central z-x plane) characteristics for the reverse magnetization process of a tetragonal box with hard magnetic properties (see the text). In this figure, *θ* is the angle between magnetic moment direction and the *z*−*axis*. (**a**–**d**) show spin configurations of the tetragonal box occurring in the reverse magnetization process, as indicated in the hysteresis loop (**e**).

**Table 1 materials-13-03696-t001:** Summary of the scaling rules.

Parameter	Scaling Rules
Distance	*r*’ = *r*·*n*
Spin (magnetic moment)	***S***’ = ***S***·*n*^3^
Exchange integral parameter	*J*’ = *J*·*n^−4^*
Anisotropy constant in volume	*K*’ = *K*·*n*^3^
Anisotropy constant on surface	*K*’ = *K*·*n*^2^
Dipolar constant	*D*’ = *D*

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
