# Peer review of "From Atomic Level to Large-Scale Monte Carlo Magnetic Simulations"

_materials, 2020, doi:10.3390/ma13173696_

Round 1
Reviewer 1 Report
In this paper, the authors reported the finite element MC method for large-scale magnetic system simulations. The scaling rules are proposes combined with small amount of system nodes. This idea is interesting and the paper is well-organized and easy to follow.
- Line 61, remove the extra punctuation mark in sentence ‘Here we present only a general idea.based on modification’.
- Line 83, ‘curry out’ —> ‘carry out’
- Line 101, line 205, ‘Curried out’ —> ‘Carried out’
- Line 142, line 143, ‘by curry out’ —> ‘by carrying out’
- Line 116, ‘using eq. (1) using the scaling rules’ —> ‘using eq. (1) and the scaling rules’
- Lines 161-162, line 177, please correct the expressions of the units and numbers, some numbers should be superscript.
- Figure 5, add scale bar.
- In section 3, line 161, the authors reported the anisotropy K=0 for soft magnet, I assume that is magnetic crystalline anisotropy. How about the shape anisotropy, does the simulation take shape anisotropy into consideration?
Author Response
In this paper, the authors reported the finite element MC method for large-scale magnetic system simulations. The scaling rules are proposes combined with small amount of system nodes. This idea is interesting and the paper is well-organized and easy to follow.
- Line 61, remove the extra punctuation mark in sentence ‘Here we present only a general idea.based on modification’.
- Line 83, ‘curry out’ —> ‘carry out’
- Line 101, line 205, ‘Curried out’ —> ‘Carried out’
- Line 142, line 143, ‘by curry out’ —> ‘by carrying out’
- Line 116, ‘using eq. (1) using the scaling rules’ —> ‘using eq. (1) and the scaling rules’
- Lines 161-162, line 177, please correct the expressions of the units and numbers, some numbers should be superscript.
- Figure 5, add scale bar.
- In section 3, line 161, the authors reported the anisotropy K=0 for soft magnet, I assume that is magnetic crystalline anisotropy. How about the shape anisotropy, does the simulation take shape anisotropy into consideration?
Our answer.
The Authors thank for constructive review and suggestions. All the typos and figure modification have been corrected in the revised version.
According to the anisotropy, the referee is right - the K parameter reflects the magnetocrystalline anisotropy. In the present paper we have not include surface (and in a consequence shape) anisotropy into consideration. However, it can be done in the stage of a system building by attributing specific K (value and direction) to the nodes placed on a surface of the built object.
Reviewer 2 Report
The manuscript is relatively well written. However, the overall figure quality is bad. The quality of the figures 2-5 needs to improve or enlarge where possible to assist the readers. Random grammatical errors are present in the text and these should be corrected. Borrowed equations should be appropriately cited in the text. The results are based on purely simulation and theory. In practice, experimental results vary widely with the simulation results in some cases.
Author Response
The manuscript is relatively well written. However, the overall figure quality is bad. The quality of the figures 2-5 needs to improve or enlarge where possible to assist the readers. Random grammatical errors are present in the text and these should be corrected. Borrowed equations should be appropriately cited in the text. The results are based on purely simulation and theory. In practice, experimental results vary widely with the simulation results in some cases.
Our answer.
The Authors thank for constructive review and suggestions. The indicated bad quality of the figures have been improved together with language correction made by a native speaker.
In our opinion the sequence of magnetic structures obtained with increasing n (i.e. single domain -> vortex -> multi-domains magnetic structures) is experimentally observed and expected which, we are convinced, confirms correctness of the presented approach.
Reviewer 3 Report
The paper "From atomic level to large-scale Monte Carlo magnetic simulations"
by Artur Chrobak et al demonstrates how the scaling rules within the Monte
Carlo magnetic simulations for large-scale systems can be used to perform
the analysis of mesoscopic objects with a small amount of system nodes. Authors propose the scaling rules, applied to the disorder-based cluster Monte Carlo algorithm. The proposed method can be consider as a suitable tool for designing new magnetic materials
I think the manuscript can be published in Materials after some improvements of the manuscript (listed below) and after the following small recommendations are considered by the authors.
1) English has to be improved. The quality of the English, unfortunately, does not help the reader to understand the most difficult passages in the text.
I suggest the author to ask for help to a native English speaker colleague or similar.
2) The Introduction focused mainly on the methods of numerical study of magnetic materials. Since the author cite the Ref[7]-[9] describing the soft and hard magnets , it will be more instructive if spin-dynamics is also highlighted in the introduction and[Journal of Communications Technology and Electronics, 2014, Vol. 59, No. 9, pp. 914–919], [Appl. Phys. Lett. 112, 142402 (2018)] and [JETP Letters, 2018, Vol. 107, No. 1, pp. 25–29] are contained in the reference list along with Ref. 7-9.
2) (line 79 and other ) In the description of parameters the indices is missing probably and the overall mathematical formatting should be improved.
3) The proposed scaling method works if the super-cell is magnetically uniform.
This should be really explained what is the limiting size of the supercell and the scaling factor for the method im terms e.g. of exchange and/or dipolar coupling between separated spins.
4) Have the authors tested their algorithm not only for n=1,10,50,100, where is the limit?
5) The Authors stated that the proposed method can be consider as a suitable tool for designing new magnetic materials. Also it is stated in the Abstract that the proposed method could show a way to include low-level or first principle calculations to a finite element MC magnetic simulations. How this is described in the main text of the manucsciprt? Maybe add some comments on this in the discussion or, e.g. the part with the hysteresis loop simulation?
Author Response
The Authors thank for constructive review and suggestions.
The paper "From atomic level to large-scale Monte Carlo magnetic simulations"
by Artur Chrobak et al demonstrates how the scaling rules within the Monte
Carlo magnetic simulations for large-scale systems can be used to perform
the analysis of mesoscopic objects with a small amount of system nodes. Authors propose the scaling rules, applied to the disorder-based cluster Monte Carlo algorithm. The proposed method can be consider as a suitable tool for designing new magnetic materials
I think the manuscript can be published in Materials after some improvements of the manuscript (listed below) and after the following small recommendations are considered by the authors.
English has to be improved. The quality of the English, unfortunately, does not help the reader to understand the most difficult passages in the text.
I suggest the author to ask for help to a native English speaker colleague or similar.
Our answer. The text have been revised by a native speaker.
The Introduction focused mainly on the methods of numerical study of magnetic materials. Since the author cite the Ref[7]-[9] describing the soft and hard magnets , it will be more instructive if spin-dynamics is also highlighted in the introduction and[Journal of Communications Technology and Electronics, 2014, Vol. 59, No. 9, pp. 914–919], [Appl. Phys. Lett. 112, 142402 (2018)] and [JETP Letters, 2018, Vol. 107, No. 1, pp. 25–29] are contained in the reference list along with Ref. 7-9.
Our answer. We agree, the suggested references were included in the revised version.
(line 79 and other ) In the description of parameters the indices is missing probably and the overall mathematical formatting should be improved.
Our answer. Done.
The proposed scaling method works if the super-cell is magnetically uniform.
This should be really explained what is the limiting size of the supercell and the scaling factor for the method im terms e.g. of exchange and/or dipolar coupling between separated spins.
Our answer. The natural limit size of the super-cell for obtaining physically reliable results is the so-called critical coherence radius which is the maximum size of a uniformly magnetized particle. This value can be determined analyzing real magnetic materials (the exchange interaction energy and magnetic saturation are taken into account). Adequate text and reference have been added in the revised version.
Have the authors tested their algorithm not only for n=1,10,50,100, where is the limit?
Our answer. In the case of our examples (soft magnets and not so strong exchange energy) the limit is about 100 (we taken the internode distance r=0.28 nm, the critical radius is in order of tenth nm) We carried out the simulation for n equals 500 and even 1000, but in this way we produce nonphysical volumes with coherent rotation of the magnetic moments inside.
5) The Authors stated that the proposed method can be consider as a suitable tool for designing new magnetic materials. Also it is stated in the Abstract that the proposed method could show a way to include low-level or first principle calculations to a finite element MC magnetic simulations. How this is described in the main text of the manucsciprt? Maybe add some comments on this in the discussion or, e.g. the part with the hysteresis loop simulation?
Our answer. The connection with low-level calculation lies in possible determination of the S, J and K parameters for the unscaled system based on, for example, DFT calculations. Using these parameters one can enlarge the system utilizing the proposed scaling rules. Adequate comment was included in the revised version.
Reviewer 4 Report
This is an interesting scientific paper, well structured, publication in MDPI Materials is a logical decision.
1) Small corrections on presentation are needed:
I observe some formatting issues in the manuscript, like different fonts and styles mixture, MDPI journal paper template must be strictly adopted for better readability. All the “v”-s “q”-s are with their own font, some of U, D, L –s also differentiate from the main font [probably this is a hidden message with quark letters]. Fonts in equations symbols and in the text must be similar, same for chapter titles.
Line 52 is a part of previous line 51.
Line 61 - mistaken comma.
Figure 3, text caption, probably “regular box” you mean “square box”.
Lines 58, 92, 99, 106, wrong paragraph indention.
Line 142, and Line 205 “curry out” = carried out? Line 133, “Choice” -> “Chose”.
2) Deeper improvements are strongly recommendable:
Used solution minimization criterion must be explained better, you are mentioning “free energy” minimization, after “system energy change” and “thermal energy”. Also criterions on structural mesh generations must be explained, how to connect rij with “system energy change and the thermal energy”.
I fill some tension in introduction scaling factor – n (after it is enlargement factor, and further enlargement parameter, please use one), in my opinion must be introduced as a ratio n = r2/r1, now existing expression at line 82 “a/n3” looks strange.
Energy is missing in Table 1.
Add a flowchart (block diagram) of the proposed algorithm.
Not sure how boundary conditions can be applied at such scaling, probably only open boundary problems can be considered.
Error estimation on presented results is missing, adding it is very important. Results must be compared with something. Color bars on Figures 2-4 will be appreciated.
Conclusion chapter, I do not find coverage of some of author claims “Simulations can be a very useful tools for designing new magnetic materials.”, “One can observe typical magnetic structures characteristic for nano and mesoscopic soft magnetic objects.”, “It was also shown usefulness of the method for simulation magnetization processes of hard magnetic materials.”, not sure where and how “atomic properties” are used here, however authors are free to proceed for publication and prove these claims in their future work.
Author Response
The Authors thank for constructive review and suggestions.
This is an interesting scientific paper, well structured, publication in MDPI Materials is a logical decision.
1) Small corrections on presentation are needed:
I observe some formatting issues in the manuscript, like different fonts and styles mixture, MDPI journal paper template must be strictly adopted for better readability. All the “v”-s “q”-s are with their own font, some of U, D, L –s also differentiate from the main font [probably this is a hidden message with quark letters]. Fonts in equations symbols and in the text must be similar, same for chapter titles.
Line 52 is a part of previous line 51.
Line 61 - mistaken comma.
Figure 3, text caption, probably “regular box” you mean “square box”.
Lines 58, 92, 99, 106, wrong paragraph indention.
Line 142, and Line 205 “curry out” = carried out? Line 133, “Choice” -> “Chose”.
Our answer. All the above comment were taken into account.
2) Deeper improvements are strongly recommendable:
Used solution minimization criterion must be explained better, you are mentioning “free energy” minimization, after “system energy change” and “thermal energy”. Also criterions on structural mesh generations must be explained, how to connect rij with “system energy change and the thermal energy”.
Our answer. The minimization criterion is well explained in our previous work (ref [22] in the revised version). It is to reach thermodynamic balance (at a specific temperature) which means minimization of the system free energy (i.e. the system energy plus the entropic term) In order to omit some repetitions we proposed the readers to read the paper when the disorder-based cluster MC method is introduced. The internode distance rij has an influence on the system energy only by the dipolar interactions, as shown in eq. 1.
I fill some tension in introduction scaling factor – n (after it is enlargement factor, and further enlargement parameter, please use one), in my opinion must be introduced as a ratio n = r2/r1, now existing expression at line 82 “a/n3” looks strange.
Our answer. Good idea, thank you. In the revised text, the definition of the n parameter were explained as suggested . The 82 line was rebuilt.
Energy is missing in Table 1.
Our answer. Energy of the system is not scaled by the n parameter directly. Therefore, in our opinion it is not necessary to include this quantity in Table 1.
Add a flowchart (block diagram) of the proposed algorithm.
Our answer. Done.
Not sure how boundary conditions can be applied at such scaling, probably only open boundary problems can be considered.
Our answer. It depends on the intention. The system can have periodic boundary conditions, making it infinite. However, the achievement (we hope) of the presented method is the possibility to simulate “single” objects of mesoscopic size.
Error estimation on presented results is missing, adding it is very important. Results must be compared with something. Color bars on Figures 2-4 will be appreciated.
Our answer. The error of the obtained spin configuration has marginal meaning accounting relatively low temperature. At the end of iteration steps the energy of the system as well as directions of spins vary in the sixth-seventh digit. In a consequence, the determined magnetization error is in the same level.
Conclusion chapter, I do not find coverage of some of author claims “Simulations can be a very useful tools for designing new magnetic materials.”, “One can observe typical magnetic structures characteristic for nano and mesoscopic soft magnetic objects.”, “It was also shown usefulness of the method for simulation magnetization processes of hard magnetic materials.”, not sure where and how “atomic properties” are used here, however authors are free to proceed for publication and prove these claims in their future work.
Our answer. The aspect of including atomic properties lies in the possibility to determination the S, J and K parameters using, for example DFT calculation. Furthermore, these quantities can be used for analysis of the large rescaled system.